# Identification of Microorganisms Using an EWOD System

**DOI:** 10.3390/mi13020189

**Published:** 2022-01-26

**Authors:** Jung-Cheng Su, Yi-Ju Liu, Da-Jeng Yao

**Affiliations:** 1Institute of NanoEngineering and MicroSystems, National Tsing Hua University, Hsinchu 300, Taiwan; sue.ivy.ping@gmail.com; 2Food Industry Research and Development Institute, Hsinchu 300, Taiwan; Liuyiju0422@hotmail.com; 3Department of Power Mechanical Engineering, National Tsing Hua University, Hsinchu 300, Taiwan

**Keywords:** microorganism identification, electrowetting on dielectric system, digital microfluidics system

## Abstract

Among the advantages of an electrowetting-on-dielectric (EWOD) chip are its uncomplicated fabrication and low cost; one of its greatest strengths that might be applied in the field of biomedical technology is that it can accurately control volume and reduces the amount of samples and reagents. We present an EWOD for the biochemical identification of microorganisms, which is required to confirm the source of microbial contamination or quality inspection of product-added bacteria, etc. The traditional kit we used existed in the market; the detection results are judged by the pattern of color change after incubation. After a preliminary study, we confirmed that an image-processing tool (ImageJ) provides a suitable method of analysis, and that, when the concentration of the sugar reagent is 38 µg/µL, the best operating parameters for the EWOD chip in silicone oil are 40 V and 1.5 kHz. Additionally, we completed the biochemical identification of five bacterial species on the EWOD chip at the required concentration of the kit. Next, we found a decreased duration of reaction and that the least number of bacteria that were identifiable on the chip lies between 100 and 1000 CFU per droplet. Because the number of bacteria required on the chip is much smaller than for the kit, we tested whether a single colony can be used for identification, which provided a positive result. Finally, we designed an experimental flow to simulate an actual sample in an unclean environment, in which we divided the various processed samples into four groups to conduct experiments on the chip.

## 1. Introduction

Since the emergence of a microfluidic system in the 1950s, much research has been devoted to their development. It is now possible to integrate the experimental procedures that require a large laboratory space and tedious operations by technicians on a small chip, that is, a lab on a chip (LOC); because of its various advantages, microfluidic chips are a key area of interest in the field of biotechnology [1]. According to Haeberle and Zengerle, the mechanism of a microfluidic system applied to the laboratory chip can be distinguished across many types, among which is digital microfluidics (DMF). The technology that we applied in this research is one type of DMF, electrowetting-on-dielectric (EWOD), which involves adjusting the wetting properties of a material surface on applying a voltage [2]. The advantages of using an EWOD system to drive the droplets are the decreased risk of contamination and decreased sample volume [3]. In addition, operation with complicated geometric structures or external equipment, such as injection pumps or valves, is unnecessary because the operating mechanism is a discrete droplet mode. In 2003, Kim et al. published a study describing their application of digital circuits to construct EWOD chips and completed four basic fluid operations—creating, transporting, cutting and merging, and found the important parameters for the creating and cutting of droplets [4]. According to the above research and advantages, such a system has the potential to develop into point of care, and gave rise to lab on a chip. The EWOD system is, therefore, widely used in such fields of biomedicine as DNA detection and extraction [5,6,7], immunoassay [8,9], reproductive medicine [10,11] and microbial detection [12,13].

Microorganism identification is required for the confirmation of the source of microbial contamination, quality inspection of product-added bacteria, and quality control of standard strains, etc. Thus, there is plenty of research invested in improving the method of bacterial detection [14,15]. In this research, we perform the identification based on the biochemical principle. Each genus of bacteria has a characteristic protein expression. Many bacterial proteins are similar, but some unique biochemical pathways define each bacterial genus; even the species within a genus can express varied proteins. These differences in protein expression between bacteria can be utilized for the identification of specific bacteria. So far, three phenotypic methods for bacterial identification have been developed, including biochemical testing, chromogenic media and matrix assisted laser desorption–ionization time of flight mass spectrometry (MALDI-TOF MS) [16]. The commercially available kit that we used, the analytical profile index (API), is based on biochemical principles and distinguishes Enterobacteriaceae of various kinds by several types of sugar. A major drawback of biochemical testing is the duration taken to detection, which is typically 24–30 h, as this action relies upon a period of incubation to produce sufficient bacterial colonies for a minimum detectable concentration of this kit, 1 McFarland (McF), which is about 3 × 10^5^ CFU/µL and the reaction period for changing color.

Because of the advantages of an EWOD system in the biochemical field, we expect to integrate the EWOD technology and the commercial kit to improve the bacterial identification based on biochemical principles, as seen in Figure 1. One advantage of an EWOD system is the convenience of the electrode design, because the electrode arrays required for varied experiments can be made through fundamental lithography; therefore, we can easily design a mask of electrode arrays to identify two strains at the same time on the chip. Moreover, as an EWOD drives droplets on applying a voltage to cause a pressure difference in the droplets, we only need to add the samples or reagents in the reservoirs and to control the electrode to simplify the steps. A major advantage of an EWOD is diminishing the volume of a sample to around 10 nL, which results in an increased ratio of surface to volume. Furthermore, we expect that we can decrease both the number of bacteria required and the duration of reaction. This research consequently provides an easier method for the identification of microorganisms based on biochemical methods using an EWOD system.

## 2. Materials and Methods

### 2.1. EWOD System

The basic principle of EWOD is that an externally added electrostatic charge may modify the capillary forces at an interface, even if the electrode is covered with a thin film. When an electric field is applied, the electric charge accumulates in the liquid/solid interface, thus decreasing the corresponding interfacial energy in the dielectric surface, causing changes in surface wettability and contact angle of the droplet [4]. According to the above principle, operating samples can be performed by applying a voltage in the direction of movement, so that the one-side contact angle of the droplet changes and induces the pressure difference in the liquid to drive the droplet movement.

The main architecture of the EWOD system in our lab is divided into a channel and an electrical signal for control. The electrical signal is provided by a function generator (33220A, Keysight, Santa Rosa, CA, USA) and a power amplifier (A304, A. A. Lab System Ltd., Ramat-Gan, Israel). The signal generated by the function generator and magnified by the amplifier is sent to the circuit, which is connected to a relay board. The channel control is from the command PXI-6512 (National Instruments Corp., Austin, TX, USA) via a program (LabView) used to switch the relay on the relay board, allowing the signal to pass accurately through the wires on a clamp (CCNL050–47-FRC), which is connected to the chip.

### 2.2. Chip Design and Fabrication

The size of a single chip is 78 mm × 28 mm. The chip contains reservoirs, control electrodes and reaction electrodes—the dimensions for which are shown in Figure 2a. Because the number of switches in our EWOD system is limited to 64, we designed some electrodes to be controlled with the same switch, and only one electrode way for two distinct bacterial species. We thus used alcohol and deionized water to clean the way of the control electrodes before loading a second strain, as shown in Figure 2b. Loading the first bacterial solution in the left side reservoir is first operational step of identification of bacteria on the EWOD chip, and switches the corresponding electrodes by LabView to generate six bacterial droplets and move each of them to upper-reaction electrodes. As mentioned above, we used alcohol and deionized water to disinfect the movement path for the bacterial solution. In the third step, another bacterial solution was added and the first step was repeated, but each droplet was moved to the lower reaction electrodes. Finally, as illustrated in Figure 2b, the six reaction reagents were generated by voltage control, moved to the corresponding reaction electrodes and placed in different reaction electrodes to mix with the bacterial solution and analyze the color change through ImageJ at 16 h.

The controlling electrodes are made of indium tin oxide (ITO), coated on glass because of its two main properties--electrical conductivity and optical transparency. For detailed information about the fabrication, we refer to article [17]. For the bottom plates, after standard cleaning, the ITO glass was spin-coated with photoresist (AZ5214) followed by exposure to a dose of 60 mJ/cm^2^, development and etching. After removal of the photoresist, 1.0 μm thick SiO_2_ was coated as a dielectric layer with plasma-enhanced chemical-vapor deposition (PECVD). The next step was coating a liquid material, spin-on glass (SOG), which was baked at 200 °C for 120 min, so as to fill the pinholes on a dielectric layer to increase the stability of EWOD devices for the manipulation of biomedical droplets [18]. The top plate, coated with ITO, serves as a ground electrode. Cytop was spin-coated by 3000 rpm and 30 s on both plates for hydrophobic purposes, and dried at 180 °C 30 min after coating. Figure 3 represents the EWOD chip after fabrication.

### 2.3. Sources of Bacteria in the Experiments

The strains used in this experiment were all obtained from Food Industry Research and Development Institute (FIRDI) by searching the website of Bioresource Collection and Research Center (BCRC). The five strains are *Escherichia coli (E. coli)*, *Kluyvera intermedia (K. intermedia)*, *Buttiauxella agrestis (B. agrestis)*, *Serratia odorifera (S. odorifera)* and *Edwardsiella hoshinae (E. hoshinea)*, with BCRC encodings 10362, 14808, 12221, 12223 and 16004, respectively, which are all Enterobacteriaceae. The strains selected for experimental safety are of biosafety level BSL1 and not pathogenic. Enterobacteriaceae are commonly found in contaminated water and food and might lead to opportunistic infections [19,20,21,22,23].

### 2.4. Concentration of Bacterial Solution Wrt to Absorbance Value

The concentration of the bacterial solution used in accordance with the provisions of the kit was 1 McF (McFarland standard). In microbiology, the McFarlane standard is a reference to adjust the turbidity of a bacterial suspension. According to the literature, the number of bacteria contained in 1 McF is around 3 × 10^8^/mL [20], that is, 1 McF ≅ 3 × 10^8^/mL = 3 × 10^5^/µL. To fix the concentration of the bacterial solution in the subsequent experiments, we designed an experiment to determine the relationship between the concentration of the bacterial solution and the absorbance, from which we obtained a table of absorbance value and bacterial solution. We scraped several colonies into sterile water and measured the absorbance of bacterial solution for fixing the concentration. Hence, we confirm that the experimental results are affected by different concentrations.

The concentration of different strains was estimated using the coating-plate method and corresponded to the measured spectrophotometric value. However, a slight concentration difference does exist, but all concentrations are close to 1 McF ≅ 3 × 108/mL.

### 2.5. Analytical Profile Index (API)

In this experiment, we used the API kit in a biochemical characteristics test system for Enterobacteriaceae (BioMérieux, Marcy-l'Étoile, France). The test strip consists of 20 plastic tubes (cupules) with dehydrated reagents. When a bacterial solution is added to the cupules, after a period of culturing (16–24 h), the bacterial metabolites alter the color so as to distinguish various bacterial species. Because of the limited number of electrodes in our EWOD system, we decided to use six reagents in the kit to distinguish five strains. The selected reagents are all sugars, namely glucose (GLU), D-mannitol (MAN), inositol (INO), D-sorbitol (SOR), D-sucrose (SAC) and amygdalin (AMY), and contain a pH indicator. An appropriate amount of sterilized water was added to the corresponding cupules, and a micropipette was used to transfer it to the labeled 1.5-mL tube after 30 min.

These six reagents were selected because there is a common biochemical metabolic pathway in Enterobacteriaceae, which is called mixed acid fermentation [24]. This path serves to metabolize and to decompose sugars, such as glucose, and to produce various acid products, which turn the reagents from blue to yellow because of the acid-base indicator. As bacterial species metabolize varied sugars, the color of the reagent after reaction can serve to discriminate between the bacterial species.

## 3. Results and Discussion

### 3.1. Preliminary Study

We used sterilized DI water (50 µL) to redissolve the reagents, and referred to the label in the kit to obtain the reagents with concentration 38 µg/µL. The six reagents were tested for creating, transporting, cutting and merging in air or an oil bath on the EWOD chip. The experiments showed that the lowest voltage for the formation of droplets in air is greater than in the oil bath. The higher the operating voltage, the greater the rate of chip damage. Furthermore, driving the droplets in the silicon oil environment can decrease the rate of evaporation of the reagents. Considering that the subsequent reagents must be cultured after mixing with the bacterial liquid for at least 16 h, we decided to operate in an oil bath afterwards. The test showed that the best operating voltage and frequency are 40 V and 1.5 kHz, which can move the droplets smoothly without causing dielectric breakdown.

Finding a method to detect a color change came next. After the biochemical reagent and the bacterial solution were mixed and cultured for several hours, the bacteria metabolized the sugar in the reagent and produced acidic products, which changed the color of the mixture from blue to yellow because of an indicator in the reagents. We discovered that using an image-processing tool (ImageJ) to analyze the hue of the mixture photograph was a method that has been used elsewhere [12,25]. The hue value, 0~360 in total, refers to a basic property of color, whereby each value represents a color. As this experiment involves a process from blue to yellow, the expected hue is from around 210 to 60.

After completing the abovementioned tests, we used a bacterial solution with a concentration of 10^5^ CFU/µL, to identify the microorganism on the EWOD chip. To make the color change of the reaction more obvious, we mixed the bacterial liquid and the reagents in volume ratio of 1:2, and allowed mixtures to stand for 18 h, which is the interval required according to a description in the kit. If the color of mixtures alters from blue to yellow, it means that the bacteria uses this sugar to generate acid products, which decreases the pH and causes a color change through the pH indicator. Photographs of cultivated *E. coli* with six reagents are shown in Figure 4a. After that, we analyzed (with ImageJ) the hue of the mixtures, and the experiment was performed in triplicate for each of the five strains, as shown in Figure 4. The hues of the mixtures, which show a positive reaction, decreased from about 200 to 60. Comparing the experimental results with Table 1, provided by the kit, we confirmed that bacteria can still grow and perform coloration reactions in a small volume, and that the Enterobacteriaceae of various kinds could be identified with specific patterns of colors.

### 3.2. Advantage of Integration of EWOD and API

The major advantage of an EWOD chip is that it can accurately control the volume of the droplets and reduce the amount of sample used, which yields an increased ratio of surface to volume; therefore, we expect that the reaction on the chip is more rapid. Furthermore, we used *E. coli* to conduct experiments and compared the fastest period of completion on the chip with the commercial kits through the records at various time points. The results, presented in Figure 5a, demonstrate that the complete reaction of *E. coli* takes 16 h on the kit and about 10 h on the EWOD chip. These results confirmed that the outcome can be obtained more rapidly on the chip at the same concentration.

An increased ratio of surface to volume not only reduces the reaction time, but might also reduce the number of bacteria required for bacterial identification. To verify this expectation, we diluted the bacterial solutions and mixed with six kinds of sugars on the chips. The results of the numbers of *E. coli* are shown in Figure 5b,c. We found that the smallest number for a positive reaction is 314 CFU. After testing the varied concentration of bacterial solutions and culturing for 18 h, the lowest number of each of the five strains (the lowest concentration Volume for reaction) are shown in Table 2, which also demonstrates the number of bacteria required on the API kit and the chip after calculation. As the number of bacteria required for the chip is much less than for the kit, we considered using the EWOD chip to identify a single colony. First, we found a relationship between the area of the colony(mm^2^) and the number of bacteria. As a result, when using a concentration of 8 × 10^3^ CFU/µL of *E. coli,* which when calculated from Table 2, is (4.3 × 10^5^) × 1/50, and when using 100 µL to dissolve, we required 8 × 10^5^ CFU as the number of bacteria. We then used the chart shown in Figure 5d to find the corresponding area. After dissolving a single colony, we obtained the bacterial solution and performed the experiments on the chips. The result, shown in Figure 5e, confirmed that a single colony can indeed be identified on the EWOD chip through the change of hue for a positive reaction after being cultured for 18 h. Because we completed an identification of a bacterial species on the chip with only a single colony, we were able to reduce the time needed to enlarge the colony, which is about 24 h.

### 3.3. Simulation of Analysis of a Real Sample on an EWOD System

Next, we simulated the bacteria of a real sample in an experiment on the chip with a single colony, to confirm that the integration of the kit and the EWOD system improved the microbial identification based on biochemistry. Commercially available rice balls from a supermarket were used in subsequent experiments, because there were less preservatives, which would affect our results. Enterobacteriaceae are detectable in food if they become contaminated with unclean tools or human hands during food processing or packaging. We imitated impure rice balls by touching with a polluted cotton swab, which was wiped in a sink and the edge of a toilet, and undertook bacterial detection with the EWOD chip. As shown in Figure 6b, we divided the experimental samples into four groups through the process steps. The first group was not contaminated with filth; the second group was exposed to pollution using a dirty cotton swab; the third and fourth groups were sterilized after contact with filth, but *E. coli* was added to the latter group after sterilization. After overnight culture, we suspended each group of several isolated colonies in sterilized water and performed identification on the chips. In the designed process, we adapted one of the current SGS methods for detection of *E. coli* in food, called the Pour plate method, as shown in Figure 6a.

The patterns of color after reactions in groups 1 and 2 are different from those of the previous five strains. We used the file attached to the API kit to test the strains corresponding to the color pattern, and inferred the possible strains detected. The results of the two colonies in group 1, labelled A and B, are shown in Figure 6. The results indicate that the colony might be *Pasteurella pneumotropica* or *Burkholderia cepacian.* More colonies grew in the second group, of which we took five, numbered from 1 to 5. The result of number 1, as shown in Figure 7a, indicated that it might be *Yersinia enterocolitica* or *Pantoea spp 1*, and numbers 2 and 5 might be *Serratia marcescens* or *Serratia plymuthica*, according to the results presented in Figure 7b. Moreover, based on the results in Figure 7c, the remaining two colonies, numbered 3 and 4, might be *Enterobacter asburiae* or *Serratia ficaria*. The purpose of the third and fourth groups were to test whether the bacteria attached to the rice ball could be removed after sterilization, and whether the previously used *E. coli* would also attach to the rice ball. As the third group of samples generated no colony on the solid medium after incubation, we were certain that the bacteria were eliminated on sterilization; the fourth group of samples grew many colonies after overnight culture, and five individual colonies were picked and then detected on the chips. No data are presented because group 3 did not grow any colonies, and the experimental results of the five colonies selected by group 4 were consistent with the initial results. As expected, all results of the selected colony are the same as the previous results of detecting *E. coli*. We found that the *E. coli* strain that we selected also attached to the rice ball.

Using the results presented above, we confirmed that the integrated use of EWOD and API can indeed detect a single unknown colony. We found that it is unnecessary to enlarge an isolated colony by placing it on another solid medium and culturing overnight. Utilizing the EWOD system can thus decrease the time needed and simplify the steps required for bacterial identification, as shown in Figure 8.

## 4. Conclusions

The advantages of an EWOD system in the biochemical field include the simplicity of electrode design, a precise drop volume control through applying a voltage and a decreased sample volume. In this article, we simplified the process of bacterial identification by combining a biochemical test with an EWOD system. Re-dissolving the reagents in the API into sterilized water and examining the best parameters for manipulating on the EWOD chips was the subject of the first preliminary study. We found the conditions needed to drive reagents smoothly are 40 V and 1.5 kHz; furthermore, the results confirmed that bio-reagent droplets can be operated on an EWOD chip without causing inactivation. In addition, the decreased usage of sample and reagent volume, an advantage of using an EWOD, increases the ratio of surface to volume. The reaction duration and the required number of the bacteria were successfully reduced, as demonstrated by the abovementioned results. Conclusively, if there are multiple unknown bacterial species in a pre-culture sample, a single colony can be used for the identification on the EWOD chip without the need to apply a plate to enlarge a single colony again, which shortens the time and simplifies the steps required to identify a bacterial species.

## Figures and Tables

**Figure 1 micromachines-13-00189-f001:**
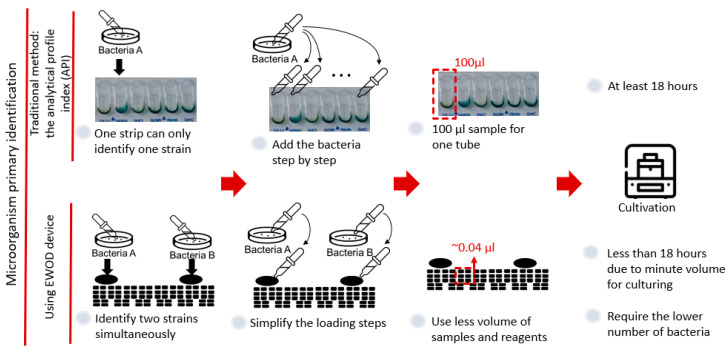
Advantage of integrating an EWOD with a commercial kit based on biochemical principles.

**Figure 2 micromachines-13-00189-f002:**
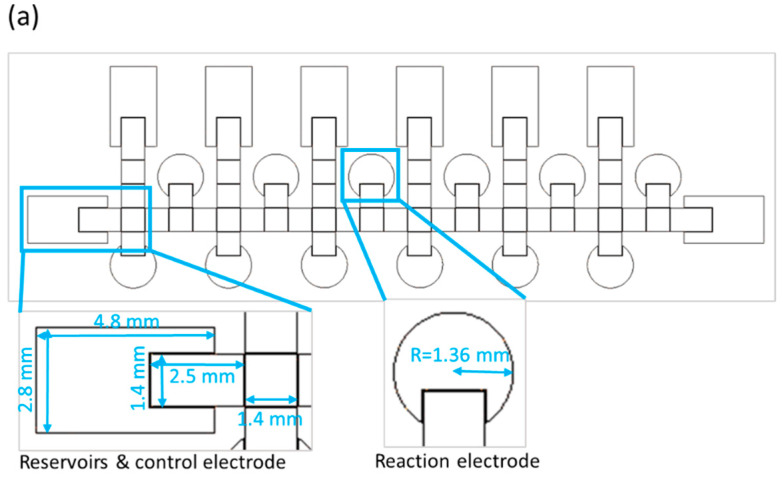
Design of an EWOD chip to detect two strains simultaneously. (**a**) schematic diagram of a parallel-plate EWOD chip; (**b**) path design for bacterial identification on the chip.

**Figure 3 micromachines-13-00189-f003:**
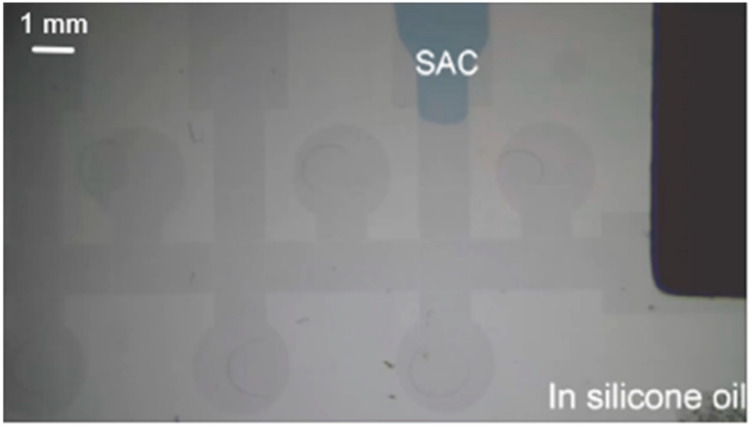
Image of EWOD chip under experimental stage.

**Figure 4 micromachines-13-00189-f004:**
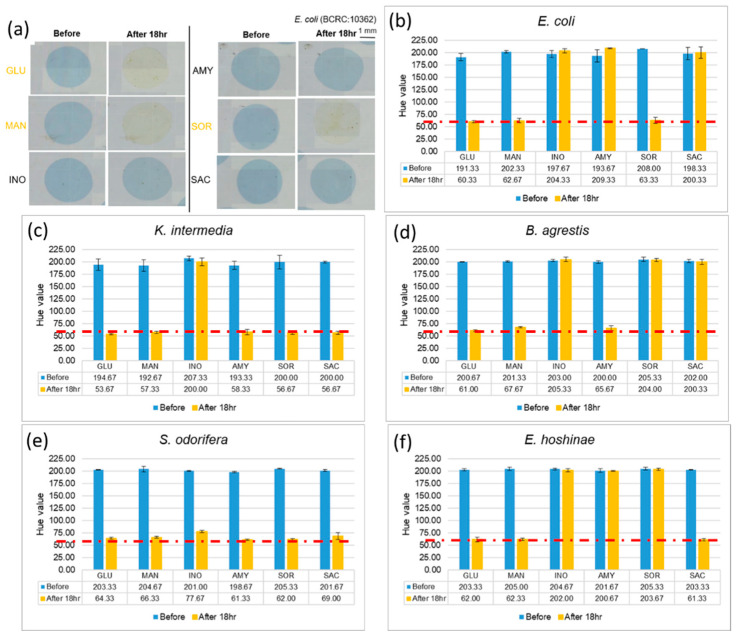
Results of cultivating selected Enterobacteriaceae at concentration 10^5^ CFU/µL. (**a**) Comparing *E. coli* images before and after culturing. (**b**–**f**) Changes of hue after 18 h indicate that different bacteria use different sugars. (Red dotted line indicated the threshold line at Hue value equal to 60).

**Figure 5 micromachines-13-00189-f005:**
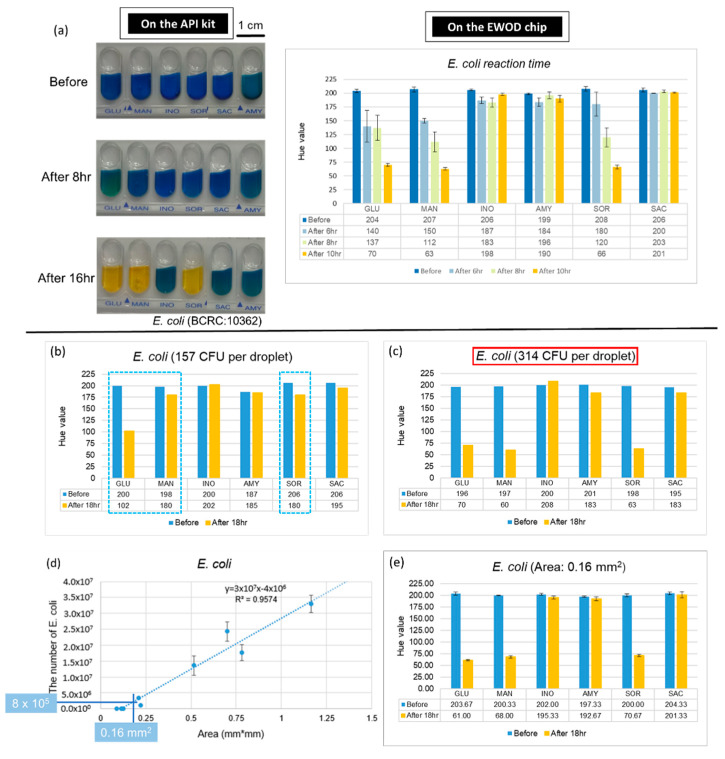
Advantages of combining API with an EWOD technique. (**a**) Reaction duration required on the kit and the chip. (**b**,**c**) Data of number of *E. coli* for identification. Dotted line indicates that the Hue value of these three mixtures, which supposed be positive reaction, did not approach 60. (**d**) Number of bacteria in a single colony. (**e**) Results of *E. coli* detection with single colony.

**Figure 6 micromachines-13-00189-f006:**
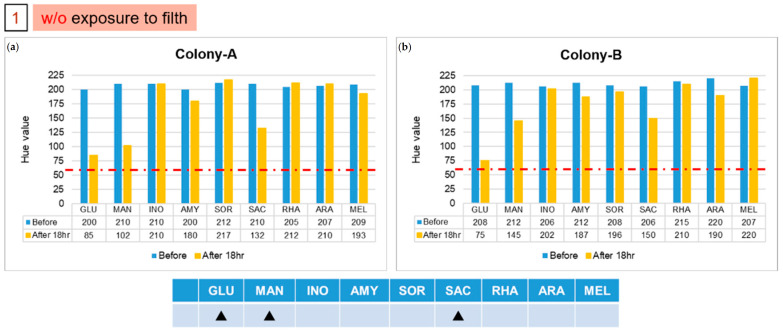
Results of two isolated colonies, (**a**) Colony-A and (**b**) Colony-B, in the first group with experiments on chips. The solid triangle symbol indicates that this unknown strain does not use the corresponding sugar in large quantities, so the hue would not change to 60. (Red dotted line indicated the threshold line at Hue value equal to 60).

**Figure 7 micromachines-13-00189-f007:**
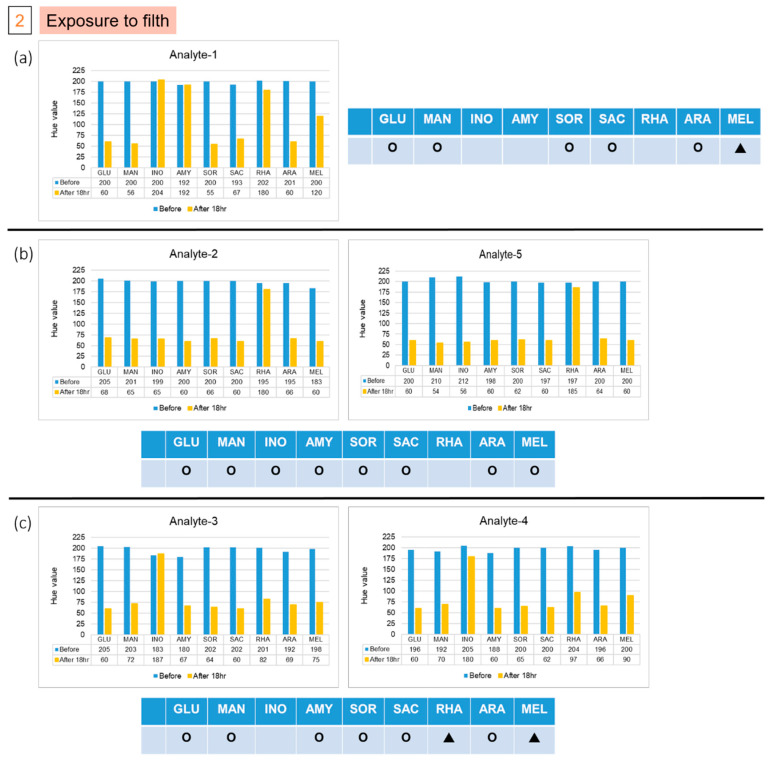
Results of five isolated colonies in the second group, (**a**) Analyte-1, (**b**) Analyte-2 and Analyte-5, and (**c**) Analyte-3 and Analyte-4, which was in contact with filth, from the experiments on the chips. (Hollow circles means Negative; filled triangles means Positive; and Red dotted line indicated the threshold line at Hue value equal to 60).

**Figure 8 micromachines-13-00189-f008:**
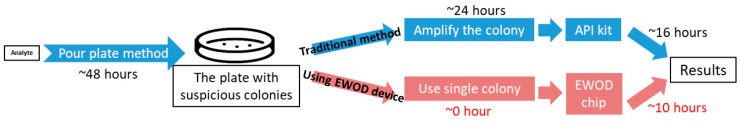
Illustration of the duration difference between doing the bacterial detection in a traditional kit and on the chip.

**Table 1 micromachines-13-00189-t001:** Reference results of five Enterobacteriaceae provided by API: Mark(V) indicates that the corresponding bacteria can metabolize the corresponding sugar and cause the mixture to change color after 18 h.

O: Positive Rection	GLU	MAN	INO	AMY	SOR	SAC
*E. coli*	V	V			V	V
*K. intermedia*	V	V		V	V	
*B. agrestis*	V	V		V		
*S. Odorifera*	V	V	V	V	V	V
*E. hoshinae*	V	V				V

**Table 2 micromachines-13-00189-t002:** Result of the calculation shows the number of five strains required on the API kit and the chip. The field with the lowest concentration was obtained from our experiment and “50×” indicates that the required concentration on the chip is one-fifth the time on the kit.

Bacteria	Concentration (CFU/μL) Required for Kit	The Lowest Concentration (CFU/μL)	Volume for Rection (μL)	The Lowest Number of Bacteria for Rection (CFU)	$\frac{\mathrm{API}\;\mathrm{kit}}{\mathrm{EWOD}\;\mathrm{chip}}$
*E. coli*	$4.3\times 10^{5}$	50×	~0.04	314	$9.4\times 10^{4}$
*K. intermedia*	$4.5\times 10^{5}$	20×		882	$3.4\times 10^{4}$
*B. agrestis*	$7.1\times 10^{5}$	50×		556	$5.4\times 10^{4}$
*S. odorifera*	$1.8\times 10^{5}$	50×		355	$8.5\times 10^{4}$
*E. hoshinae*	$1.6\times 10^{5}$	10×		627	$4.8\times 10^{4}$

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
