# Peer review of "Identification of Microorganisms Using an EWOD System"

_micromachines, 2022, doi:10.3390/mi13020189_

Round 1

Reviewer 1 Report

The authors present a biochemical identification of microorganisms by using an Electrowetting On Dielectric (EWOD) system. In particular, they use EWOD in order to manipulate and reduce the volume of the sample droplets. The manuscript is well written and I believe would be interesting for the readers of Micromachines.

I would like only to indicate one minor issue, as listed below:

Line 90: The authors say that: "When an electric field is applied, the charge changes free energy of the dielectric surface, causing changes in surface wettability and contact angle of the droplet [4]."

I don't agree with the above sentence. In my opinion, a more accurate definition of EW phenonemon would be: "Electric charge accumulates in the liquid/solid interface thus decreasing the corresponding interfacial energy"... causing changes in the apparent contact angle of the droplet

See ref:

Mugele, F., & Baret, J.-C. (2005). Electrowetting: from basics to applications. In Journal of Physics: Condensed Matter (Vol. 17, pp. R705–R774). https://doi.org/10.1088/0953-8984/17/28/R01

for more information about EW phenomenon

Reviewer 2 Report

The authors report an Electro-wetting on Dielectric (EWOD) chip for biochemical identification of microorganisms by using Analytical profile index (API). By performing on the EWOD chip, the amount of sample and reagent, and the reaction duration is reduced. The EWOD chip has a higher sensitivity than the conventional method, can detect a single colony without overnight incubation and can identify two strains simultaneously. Despite the hard work put into the design and experiments, many parts of the paper need to be improved, especially written English. Specific comments are as follows:

  1. It is difficult to understand what the photos of the top row is showing in Figure 1.
  2. The authors should include a real image of the EWOD chip.
  3. Line 149, the sentence is incomplete, please correct it.
  4. Table 1, please correct the unit of “Concentration”.
  5. Table 1 is trivial. The authors should consider eliminating it from the manuscript, or moving it to supporting information.
  6. Please review the sentence between line 156 and 159.
  7. In line 185, what does “an oil bath” mean? Could you describe it in more details?
  8. Could you explain why you didn’t use a consistent bacteria concentration and dilution factor (example 10-fold dilution) across the manuscript? Specifically, E. coli concentration is different between Table 1 (4.3 x 105 CFU/ µL), Section 3.1 (105 CFU/µL) and Section 3.2 (314 CFU).
  9. There is no Figure 5 in the manuscript.
  10. In Figure 4, The chart (b), (c), (d), (e) should have the same dimension and be aligned.
  11. It is not sure why third and fourth groups (line 282) are added to the experiment. The purpose or the impact of sterilization of rice balls after contamination is not understandable. The authors should explain the purpose of third and fourth groups, and also how the sterilization was performed in more detail. Additionally, the difference between third and fourth is not clear, and also data of the two groups is not shown.
  12. In Figure 8, please consider changing the position of the text “Tradition method” and “Using EWOD device” since it is difficult to read.
  13. English needs to be improved.

Reviewer 3 Report

In this manuscript, authors presented an EWOD-based digital microfluidic system for the identification of microorganisms based on their metabolic profile. They demonstrated a proof-of-concept with five bacterial strains and spiked food samples. It showed that the EWOD-based identification could significantly improve the sensitivity and reduce the bacteria required for each test. The study is sound, but the data are not well presented, and the manuscript is not carefully prepared. The following issues must be addressed before the manuscript is considered for publication.

  1. The writing needs MUCH improvement. It is often quite difficult to understand what the authors mean in many places. The abstract has too many details but does not provide a synopsis of the manuscript with key findings.
  2. Authors should present pictures that show the EWOD droplet operation for the assays, not just schematics.
  3. It is very confusing to present the hue value in bar chart with different bar color. The difference in magnitude of hue value presented in the bar chart does not carry much information. Using blue and yellow to represent the hue value before and after the incubation is misleading. It makes readers believe the color after incubation always turn yellow; however, it is the magnitude of the bar that represents the color. Authors need to find another way to present the results.
  4. It becomes even more confusing with more bars of different colors in Fig. 4. The results from the benchmark experiment and the EWOD platform should be presented in the same way for easy comparison.
  5. Figure 5 is missing.
  6. In Figure 6, instead of calling “Analyte-A”, why not “Colony-A”. It is quite misleading by calling it analyte, which makes readers wonder whether it refers to the bacteria or the sugars.
  7. Column 3 of Table 3 should not carry any unit.
  8. Bacteria colony is “suspended” in the reagents, not “dissolved”.

Round 2

Reviewer 2 Report

The authors seem to have made corrections.

Reviewer 3 Report

My previous comments have been addressed. I am OK content wise.